# OpenReview forum: "FLIRT: Feedback Loop In-context Red Teaming"
_ICLR.cc/2024/Conference — Submitted to ICLR 2024_

### Official Review · Reviewer_dszm · 2023-10-23

**Soundness:** 1 poor
**Presentation:** 3 good
**Contribution:** 2 fair
**Rating:** 5
**Confidence:** 4

**Summary:**

This paper studies automatic red-teaming where new adversarial prompts are generated based on seed in-context examples and a few simple strategies (e.g., FIFO, LIFO) are explored to hot-swap the generated prompts with existing in-context prompts. Experiments are mostly conducted on text-to-image models and the major baseline is an existing stochastic few-shot method where the adversarial prompts are more random than the proposed update strategies.

**Strengths:**

- The paper is nicely written and easy to read

- Safety of generative AI is an important topic

- A lot of analysis and ablations of the proposed method are presented

**Weaknesses:**

While I appreciate the effort of the authors conducting tons of experiments and analysis, my main concerns are around the evaluation of the proposed method.

- The metric of attack effectiveness is a bit misleading, as the authors mentioned themselves "the red LM learns an effective prompt that is strong in terms of triggering the text-to-image model in unsafe generation; thus, it keeps repeating the same/similar prompts that are effective which affects diverse output generation". If the prompts remains the same all the time, does it have an attack effectiveness of 100%? It doesn't sound reasonable to penalize methods that discover more adversarial prompts (though some of the prompts are not effective). A more rigorous study would be to have some categories (e.g., sexual, violent, etc.) and see how the methods perform in different scenarios. The current metric doesn't account for that and in the proposed method there also doesn't seem to be much control of what types of prompts to generate (apart from a "diversity measure"), making the attack less oriented.

- Another related issue is, as listed in Table 9, right now only 3 seed prompts are used for evaluation, and they are rather similar in nature (either sexual or violent). A more comprehensive study with larger scale would have been more convincing that the method is generally applicable and not sensitive to / relying on the prompt engineering of the initial seed prompts (quoting from paper "hand engineered by humans"). Speaking of which, how effective are the initial prompts?

**Questions:**

- Why is FIFO already so much better than SFS? Is it mostly because of the initial prompts or the fact that only successful prompts are added?

- In text-to-text experiment, why do you use two evaluators (toxigen and perspective api) for toxicity? Did you re-calibrate the scoring threshold? Can you show some model generations at different score percentiles? In my experience, for example, there could be high false negatives in toxigen in the default setting.

---

> ### Author Response · Authors · 2023-11-15
> **Author Response to Reviewer dszm**
>
> We thank the reviewer for their feedback.
>
> Regarding the first point made by the reviewer, We see two sub-questions here that we address separately.
>
> **on attack effectiveness:** First, we agree that “attack effectiveness” metric by itself is not very useful as the model may learn to repeatedly “generate” the same/similar effective prompts. This is exactly why we report the diversity metric in addition to attack effectiveness and propose an objective where we combine attack effectiveness with the diversity metric, to force FLIRT to generate prompts that are novel and effective at the same time. In this regard, we are not quite sure about the reviewer’s statement “It doesn't sound reasonable to penalize methods that discover more adversarial prompts   (though some of the prompts are not effective)”. The diversity part of our objective assures that FLIRT generates novel prompts, and the effectiveness part of the objective ensures that the new prompts are effective against the target model. Without the latter, FLIRT would generate a very diverse set of prompts, but those prompts would not be effective against the target model, thus defying the purpose of red teaming.
>
> **on different diversity metric:** The second part of the comment is about having different categories of unsafe generation (e.g., sexual, violent), and then studying the diversity of the prompts in terms of different types of unsafe generation. This notion of category-wise diversity is  interesting. However, we believe the diversity metric we study in the paper is better for our purposes as it captures a more general notion of diversity regardless of the task/categories in hand. Furthermore, it might be hard to categorize certain prompts into specific categories in which case the proposed evaluation metric might not be robust and reliable. In addition, Figure 6 (refer to the appendix) contains some experiments on using a different diversity score by reporting Rouge-L score distributions as an additional study on the diversity of the generated prompts which should answer your concern.
>
> **On the effect of seed prompts:** As mentioned in section 3, “In addition, we perform numerous controlled experiments to better understand the effect of seed prompts and how they differ from the generated prompts in the Appendix.”, Figure 5 in the appendix has a study on this. In Figure 5 we test our framework with different sets of seed prompts each containing different number of unsafe prompts and report the attack effectiveness rate. We demonstrate that even with 1-2 unsafe examples, the framework is able to generate a large percentage of successful examples. In addition, Figure 6 shows that generations are different than the seed prompts according to their Rouge-L distributions.
>
> **Why are we better than SFS:** Both factors play a role. As shown in Figure 5, the initial prompts have an effect (as we increase the number of successful prompts the attack becomes more effective; however, note that even with 1-2 successful initial prompts, we are able to obtain high attack success rates). In addition, as shown in the main body of experiments, the way we incorporate the  feedback (successful prompts) plays a significant role (results are different in a FIFO based approach compared to LIFO, scoring, LIFO-scoring). This suggests that the feedback and the incorporation of the successful prompts also play a significant role and can change the results significantly such that each approach imposes its own trade-offs. Hence, both the initial prompts and the feedback (the way that the successful prompts are added) play a role.
>
> **On text-to-text evaluators:** We only used toxigen for text-to-text experiments and perspective api for text-to-image experiments and not text-to-text. The reasons why we did this was:
> 1. To make the distinction between text-to-text and text-to-image experiments which are disjoint experiments and the result for one does not affect the result for another.
> 2.  In text-to-image experiments our classifier was applied on the input prompt while in text-to-text experiments it was applied on the output prompt; hence, for the ease of notation in section 2.3 and clarity of writing, we used two different classifiers so that we have two different notations to make this distinction.
> 3. To show that our approach is general enough to apply to different classifiers hence we used different classifiers to showcase this capability and that we are not only limited to perspective api but a wider range of classifiers.
>
> We acknowledge that there is always the risk of false negatives/positives in using any automatic classifier and hence clearly mentioned this in our paper. In addition, we perform a dedicated experiment to this in our ablation study 5, where we did experiments in which we show that our results are robust to existing noise coming from these automatic classifiers.

---

> > ### Author Response · Authors · 2023-11-22
> >
> > Dear reviewer dszm,
> >
> > As we are reaching to the end of the discussion period, we would appreciate it if you could take a look at our response and let us know if you still have concerns. Thank you.

---

### Official Review · Reviewer_2CM2 · 2023-10-27

**Soundness:** 1 poor
**Presentation:** 3 good
**Contribution:** 2 fair
**Rating:** 5
**Confidence:** 4

**Summary:**

The authors propose the in-context learning-based red teaming method named "FLIRT" which iteratively updates demonstrations according to the feedback from the target model. The FLIRT method has 4 variations in its attack strategy as following:
- First in first out (FIFO) attack : If new prompt elicit an offensive response, remove the first exemplar in the exemplar queue and add new prompt into the exemplar queue.
- Last in first out (LIFO) attack : If new prompt elicit an offensive response, remove the first exemplar in the exemplar stack and add new prompt into the exemplar stack.
- Scoring attack : Update exemplars based on scores such as attack effectiveness, diversity, low-toxicity.
- Scoring attack + LIFO : combining scoring attack and LIFO

The empirical results show that FLIRT can discover a larger number of positive test cases, that elicit offensive responses, compared to the baseline methods.
Moreover, the authors build a benchmark dataset consisting of positive test cases.

**Strengths:**

- The idea is simple and intuitive.
- The paper is well-written and easy to understand.
- The paper contains red-teaming results for both text-to-text models and text-to-image models.
- The authors evaluate the baseline method and FLIRT with GPT-Neo as a red LM, which is much cheaper than Gopher used in [Perez et al., 2022]. It is a huge contribution for the following researchers.

**Weaknesses:**

If I understood correctly, the contribution of this paper can be listed as follows:

a. Propose in-context learning methods which is better than stochastic-few-shot of [Perez et al., 2022].

b. The proposed methods can control diversity and toxicity of generated prompts.

c. Evaluate the red team methods on not only text-to-text models but also text-to-image models.

Soundness [a]: The empirical results supporting the superiority of the proposed method seem weak.

Missing reference [b,c]: There exists a previous work named Bayesian red teaming (BRT) which controls the diversity and toxicity of generated prompts and also conducts experiments on both text-to-text and text-to-image generative models [1]. BRT controls both diversity and attack effectiveness during the generation process. Also, [1] evaluates the red teaming method when the possible inputs are restricted to non-toxic texts. In this regard, several parts of the FLIRT paper are not that new.

[1] Query-Efficient Black-Box Red Teaming via Bayesian Optimization, Lee et al., ACL 2023.

**Questions:**

- Can you show the score of other kinds of diversity metrics such as self-bleu in [Perez et al., 2022]?
- In my opinion, stochastic few-shot can operate similarly to FLIRT by adjusting the temperature. For example, if we set the temperature of stochastic few-shot to a low value, the exemplar set would be constructed by the prompts with the highest scores, which is similar to Scoring attack version of FLIRT.
- Moreover, there is an obvious trade-off between attack effectiveness and diversity in red-teaming (refer to fig 2 in [Perez et al., 2022]). However, there is only superiority in attack effectiveness according to table 1. Can you show the trade-off curve between diversity and attack-effectiveness of FLIRT? If the FLIRT's trade-off curve majorizes stochastic few-shot's trade-off curve, it would be obvious evidence of the superiority of FLIRT methods.

**Details Of Ethics Concerns:**

The paper contains several subsections discussing the ethical perspectives associated with the research.

---

> ### Author Response · Authors · 2023-11-15
> **Author Response to Reviewer 2CM2**
>
> We thank the reviewer for their comments.
>
> **On the weaknesses of our work:** We kindly ask the reviewer to provide more context/substance for their criticism of the soundness of our paper.  We believe we have provided extensive experimental results and evidence to validate our claims; please see Sec. 3.  We would appreciate if the reviewer explicitly states what aspects of our results are weak, and what additional evidence needs to be provided.
>
> We thank the reviewer for pointing to the relevant work, we have added it in the revised draft. We note that while both BRT and FLIRT are read teaming methods, the underlying technical approaches are very different. BRT uses bayesian optimization to edit and find successful examples, while FLIRT is based on iterative in context learning with a feedback loop. FLIRT is more efficient as it uses in-context learning without requiring an additional expensive training. FLIRT also does not require extensive access to datasets to be trained over as it is using in-context learning and can operate using a few seed prompts.
>
> **Comment on other kinds of diversity metrics:** Figure 6 in the appendix has some experiments using Rouge-L score. We consider the same evaluation strategy as the self-instruct paper.
>
> **On SFS operating similarly to FLIRT:** SFS is a retrieval based approach in which they retrieve examples from the pool of zero-shot examples for the few-shot generations. If the pool of zero-shot generations are not successful, regardless of the temperature value, the approach would not find successful examples. On the other hand, FLIRT (especially the scoring approach) uses a feedback loop which improves upon its few-shot demonstrations starting from only a few (three or four) demonstrations in each successful iteration. In this case, if a new generation is more successful, FLIRT will consider it as its demonstration and keep improving on it in the next iterations. This has two advantages, first we do not require a large pool of zero-shot examples to be generated to begin sampling from as we are reusing our generations as demonstrations. Second, FLIRT can improve upon its demonstrations by incorporating the scoring coming from the feedback loop to further improve its demonstrations and hence the generations.
>
> **On trade-off curve:** We included the trade-off curve in Figure 7 of our revised paper.

---

> > ### Comment · Reviewer_2CM2 · 2023-11-20
> >
> > Thank you for your kind responses. First, I would like to explain why I initially scored soundness as low.
> >
> > The main results in Tables 1, 2, and 3 of the paper illustrate the diversity and attack effectiveness of the proposed methods compared to baseline methods. In Section 3.1, the first paragraph of "Attack Effectiveness" on page 5 explains that while SFS is effective at finding diverse adversarial examples, it has weaknesses in generating effective attacks. This conclusion seems counterintuitive, as the trade-off between attack effectiveness and diversity can be controlled by modifying hyperparameters like lambda 2 or temperature. For instance, we may elicit the opposite conclusion that FLIRT can generate more diverse examples than SFS by setting lambda 2 of FLIRT to 1.0 and the temperature of SFS to a near-zero value. Therefore, I believe Tables 1, 2, and 3 do not convincingly support the superiority of FLIRT over the baseline methods. This was the basis for my request for a trade-off curve.
> >
> > The trade-off curve and the authors' responses regarding the differences between SFS and FLIRT have clarified the proposed method's advantages for me, leading me to increase the overall rating. However, I suggest that the authors revise the main paper to provide more robust evidence of the proposed method's superiority.

---

> > > ### Author Response · Authors · 2023-11-21
> > >
> > > We greatly appreciate your careful reading and consideration of our response. We have revised our paper specifically section 3.1 the discussion on “Attack Effectiveness” which you had concerns about to provide more clarity. Specifically, we have incorporated your suggestions to highlight the advantages of our method based on the discussion above. We would appreciate if you could let us know whether these revisions adequately address your remaining concern,  and if so, we kindly ask you to revise the score accordingly. And if not, please provide some additional pointers that we can address to make the presentation even more robust in our final version of our paper. Thanks again for helping us to improve our paper.

---

> > > > ### Author Response · Authors · 2023-11-22
> > > >
> > > > Dear reviewer 2CM2,
> > > >
> > > > As we are reaching to the end of the discussion period, we would appreciate it if you could take a look at our response and let us know if you still have concerns. Thank you.

---

### Official Review · Reviewer_ZWk1 · 2023-10-31

**Soundness:** 4 excellent
**Presentation:** 4 excellent
**Contribution:** 2 fair
**Rating:** 8
**Confidence:** 3

**Summary:**

This work red-teams text to image and text to text models using in-context learning. They present a method called FLIRT that uses seed examples and labels from some harmful text/image classifier to help find new types of adversarial prompts with in-context learning. They test three different variations of the methods and conduct thorough ablation studies.

**Strengths:**

- Red teaming mediated by in context learning is appealing because of the inductive biases that models have and because of a human’s ability to influence the process with prompting.
- I think their dataset of 76k prompts will genuinely be useful (I haven’t personally looked through examples from it though.)
- Section 3.2. was well-done.
- Overall well-written

**Weaknesses:**

1. I get how SFS is a relevant few-show baseline. But it seems like a fairly weak one overall. Other, perhaps less-efficient baselines could have been tested. For example, one could use the type of RL-based attack technique used in [Deng et al. (2022)](https://arxiv.org/abs/2205.12548), [Perez et al. (2022)](https://arxiv.org/abs/2202.03286), and [Casper et al. (2023)](https://arxiv.org/abs/2306.09442). Other approaches based on zero-order search could also be used like [Zou et al. (2023)](https://arxiv.org/abs/2307.15043) (and several predecessor works before it). I don’t really fault the paper for not trying some of the other heavier approaches, but I think they could be discussed better.
2. Related the the above, one baseline that I do really really wish were tested is to do in-context reinforcement learning. This would be similar to the scoring attack. You could use an advanced chatbot like Llama or GPT-4, give it an appropritate prompt, start it off with any examples you’d like, and then let it learn in context from trial and error how to generate diverse adversarial prompts. Please comment on this.
3. This red-teaming strategy really seems to have the humans do most of the heavy lifting. Is it really meaningful red-teaming is it is assumed that the red team starts off with a pretty good idea of what general types of prompts trigger bad behavior? One could argue FLIRT is just a glorified data augmentation technique paired with best-of-n sampling. (Meanwhile, the in-context red teaming approach mentioned above would not involve the humans doing such heavy lifting.) Could the authors comment on how often they discovered very novel/surprising/off-distribution adversarial prompts?
4. Minor. Don’t respond to these. Only respond to 1-3.
    - Consider making the warning in the abstract red.
    - Last sentence of the abstract is vague.
    - I have never heard of red teaming being referred to before as “adversarial probing.”
    - Prior to this work, [Casper et al. (2023)](https://arxiv.org/abs/2306.09442) penalized cosine distances to get diverse samples when red teaming.
    - At no point does the paper highlight that one advantage of this red teaming technique over many others is that it’s black-box. The paper should highlight that advantage!

**Questions:**

See under weaknesses.

---

> ### Author Response · Authors · 2023-11-15
> **Author Response to Reviewer ZWk1**
>
> We thank the reviewer for the feedback.
>
> 1. We added citation and discussion around the mentioned references in our revised paper. The RL-based works in (Deng et al, Perez et al, and Casper et al) require a relatively large dataset of prompts for training. The motivation behind our work is to learn to generate adversarial prompts without access to such dataset, using a seed set of a few adversarial prompts instead. That said, one can in principle use the prompts generated via our framework (FLIRT) for doing RL, although this will not be a comparable baseline. In fact, ( Perez et al.) explored a similar approach, where they trained a red LM on data generated by their SFS approach, using both SFT and RL. From the reported results in Figure 2 from Perez et al. we observe that our approach is in par with the RL approaches, with the additional benefit of being more efficient as it does not involve training. Since efficiency is one of our primary goals, we find it reasonable and fair to compare our methods to a baseline such as SFS that require no training and is using few shot setup. In addition, Zou et al. proposes a white-box jailbreaking attacks on text-to-text models which is not fully aligned to our setup that is a black-box setup on multimodal models in the context of red-teaming.
>
> 2.  As the reviewer mentions, the proposed in-context reinforcement learning-based method sounds similar to our scoring attack, in that FLIRT uses the score as a type of feedback signal to learn effective (and diverse) prompts during an iterative learning process. Doing this with in-context RL sounds like an interesting future direction to explore. We assume the actions  and rewards would correspond to selecting prompts and effectiveness of those prompts respectively. However, it is not very clear what would be the states of the RL agent, and the corresponding policies. If the reviewer has some insights on this they could share, we’d greatly appreciate that.
>
>
> 3. We respectfully disagree with the assessment that “humans do most of the heavy lifting”. While we seed the model with a few human-provided prompts, FLIRT learns how to discover novel, effective prompts automatically. Please see Fig 6 (in the appendix) where we show how different the generated prompts are to the seed prompts according to the Rouge-L score distribution which suggests that our framework is able to discover new/novel instances. Furthermore, even if the seed prompts are not very effective, FLIRT will still learn to generate effective prompts; please see Fig. 5. where we do experiments with different sets of seed prompts that have different number of effective prompts in them and show that even with 1-2 effective seeds, the framework is able to generate various effective prompts.
>
> 4. Thank you for the great pointers. We have applied them to our revised paper.

---

> > ### Comment · Reviewer_ZWk1 · 2023-11-20
> > **Thanks**
> >
> > 1. Re: "The RL-based works in (Deng et al, Perez et al, and Casper et al) require a relatively large dataset of prompts for training." -- I do not think this is generally accurate. RL attacks are more versatile than that. If the goal was to use RL to train a policy to produce a distribution of adversarial prompt prefixes or suffixes to prompts, then you would need a dataset of prompts for training. But (1) I do not believe it would necessarily need to be relatively large. I don't know of work testing this in NLP, but universal adversarial perturbations for image classifiers are often developed over batches of just dozens of examples. And (2) a dataset of prompts would not be needed if the goal were to use RL attacks to synthesize the adversarial prompts themselves instead of adversarial prefixes or suffixes. So I think that this reply downplays the applicability of RL attacks to the same scenario as FLIRT, and I encourage the authors to double check that their modifications to the paper similarly do not downplay them.
> > 2. Re: "However, it is not very clear what would be the states of the RL agent, and the corresponding policies." The suggestion that I have is to simply write a plain-english prompt to tell a model like Llama-2 or GPT-4 that in this conversation, it should produce a prompt each turn and get a reward each turn. Then during the conversation, via trial and error (i.e. RL) it should work to develop prompts that get higher rewards.
> > 3. Could you share or point me to an example of (1) a set of prompts that you started with, (2) the best-performing set of prompts after FLIRT, and (3) the total number of examples that were generated over the course of the experiment? I would also be interested in a baseline where some non-LLM data augmentation technique (e.g. nonsynonymous word substitutions) would compare to your use of LMs for FLIRT.

---

> > > ### Author Response · Authors · 2023-11-21
> > >
> > > Thank you for the follow up discussions.
> > >
> > > 1. As you mention correctly, RL attacks are more versatile and we need to make the distinctions on what is RL used for.
> > > We have modified the paper to make this distinction clear. Specifically, we are not downplaying any of those approaches but just elaborating where each method is used. We believe that our revised version provides a satisfactory discussion of these papers, but please let us know if there are other adjustments we should make.  By requiring a relatively large dataset we meant approaches that use SFT or RL to fully train an adversarial prompt generator (e.g., in Perez et al.). FLIRT (which is an adversarial prompt generator) does not need any training hence no need for any dataset. By relative we meant 0-4 data instances (which is what is needed for FLIRT to work which can be implemented as either zero-shot or few-shot with as few examples as four or less) compared to a training/fine-tuning dataset required to train an adversarial prompt generator using SFT or RL.
> > >
> > > 2. Note that we already provide a plain-English instruction to the model (in addition to the seed prompts) describing the task; see Table 9 in the appendix. In this sense, we believe that our scoring approach in FLIRT that leverages feedback is very similar to the reviewer’s proposed RL method. One could perhaps explore different variations of the instructions (e.g., mentioning the reward explicitly per reviewer’s suggestion). However, it will be more of an extension to FLIRT rather than an established baseline.
> > >
> > > 3. (1) Tables 9, 10, and the table in the Figure 5 (in the appendix) include the seed prompts used for the text-to-image experiments, text-to-text experiments, and additional studies on the effect of seed prompts in text-to-image experiments respectively. (2) Best-performing prompts after FLIRT as per what metric and what attack strategy (FIFO, LIFO, scoring, etc)? If the reviewer means in the scoring approach the prompts that replace previous prompts that have lower scores, we can provide those. Otherwise, please let us know what best-performing means here as the goal is to generate adversarial examples that can result in unsafe image/text generation and in other approaches, such as FIFO, LIFO there is no such notion as best or worse performing as long as the prompt is successful in triggering unsafe generation. In addition, we have provided some qualitative examples in the appendix that has some examples of successful generations after applying FLIRT which can be compared to the seed prompts. (3) we ran all the experiments using 1,000 FLIRT iterations, so for each experiment the total number of examples generated is 1,000.
> > >
> > > We have included some results as per your suggestion in the revised version of the paper in the appendix (refer to section A of Appendix 5th paragraph) showing how a data augmentation technique (which augments only four set of prompts as that is only what is needed for our approach) can have limitations compared to our approach. We augment the seed prompts to create 1,000 new adversarial data points by using word substitutions, removing sentences, rewording, adding more information and sentences, and combination of these data augmentation techniques. As shown in the Rouge-L plots in Figure 7, the data augmentation technique will result in very similar prompts to that used by the seed prompts, while our approach will result in more different set of examples that have less rouge overlap with the seed prompts (notice the distribution shift from left to right which shows the amount of similarity increase for our approach compared to the baseline). This again highlights why our approach is able to generate novel instances, while a data augmentation approach is not able to. In addition, notice that our approach can work in a zero-shot setup in that case you would not even have access to the four seed examples to be able to augment them. Moreover, our approach can be customizable for any balck-box model to optimize for various objectives. Thus, stating that our approach is a data augmentation approach is not fair.

---

> > > > ### Comment · Reviewer_ZWk1 · 2023-11-23
> > > > **Raised my score to an 8, I think the paper should be accepted**
> > > >
> > > > - I think that this paper is a good step in an important research agenda. I would encourage the authors to consider pursuing the use of in-context reinforcement learning with GPT-4 or similar models.
> > > > - I appreciate the additions that have been made.
> > > > - I would encourage the authors to include some examples of the generated prompts that had the highest NSFW score in some experiment according to the NSFW detectors. Maybe these could be added to a table after tables 9 and 10.

---

> > > > > ### Author Response · Authors · 2023-11-23
> > > > >
> > > > > We are greatly appreciative of your consideration. We will for sure add your suggestions to the paper. Thanks a lot again.

---

### Official Review · Reviewer_Wisf · 2023-11-07

**Soundness:** 3 good
**Presentation:** 3 good
**Contribution:** 3 good
**Rating:** 6
**Confidence:** 4

**Summary:**

This paper proposes an automated red teaming method for text-to-image models like Stable Diffusion (with some experiments on text-to-text models as well). The method is similar to the few-shot method from Perez et al. (2022), but with several differences in design and better performance. There are extensive experiments and ablations.

**Strengths:**

- Automated red teaming is a timely and important problem, and there have been relatively few papers focusing on text-to-image red teaming
- The few-shot method from Perez et al. was an interesting approach, and I'm glad to see more exploration of this type of method
- There are many different variations of in-context red teaming explored in this paper, which could be helpful to future papers seeking to explore this space further
- The results are strong

**Weaknesses:**

- It would be good to have more baselines. E.g., methods like PEZ have also been evaluated primarily on text-to-image models, and some concurrent work from Google DeepMind would be good to compare to: https://arxiv.org/abs/2309.03409. The limited comparison to other methods is the main reason why I'm not giving a higher score initially.

**Questions:**

No questions

---

> ### Author Response · Authors · 2023-11-15
> **Author Response to Reviewer Wisf**
>
> We thank the reviewer for the feedback and the pointer to the paper. We have included a reference to the paper in our revised paper. We would like to note that according to ICLR’s guidelines “authors are not required to compare their work to concurrent work within the 2 month time frame as well as unpublished work.” This paper requested by the reviewer became public days before the ICLR submission date and is unpublished. However, we have included a citation of this work to acknowledge it in our paper.

---

> > ### Comment · Reviewer_Wisf · 2023-11-23
> > **Response**
> >
> > Thanks for adding the citations. I will keep my score as a 6.

---

### Meta-Review · Area_Chair_VUSy · 2023-12-11

**Metareview:**

The paper introduces the FLIRT method, building upon Perez et al. (2022). FLIRT maintains a list of seed in-context exemplars, iteratively updating it through a process of generating adversarial prompts, evaluating model outputs, and updating the exemplar list using strategies like FIFO, LIFO, scoring, and scoring-LIFO. The method's effectiveness is demonstrated through experiments on both text-to-text and text-to-image models. Additionally, the authors have developed a benchmark dataset for red-teaming text-to-image models.

After consideration of the significant disparities between the authors' and reviewers' perspectives, I meticulously reviewed the original paper. The paper is commendably well-written and easily understandable. I focused on the reviews by Reviewer dszm and Reviewer 2CM2, also taking into account the positive feedback from two other reviewers. However, unresolved issues remain: (1) Certain aspects of the work overlap with previous studies without achieving better performance or distinct methodological advancements. (2) The paper falls short in comparing FLIRT with a broader range of existing baselines. Moreover, the evaluation metrics used are misleading, weakening the claim of FLIRT's superiority. (3) The effectiveness of the FLIRT method is notably dependent on the initial seed prompts, indicating a sensitivity to the quality of prompt engineering.

Overall, the paper does not show novel insights in the field, and the methodological uniqueness of FLIRT is not convincingly established. Thus, the paper does not fully meet the standards expected for acceptance at ICLR.

**Justification For Why Not Higher Score:**

N/A

**Justification For Why Not Lower Score:**

N/A

---

### Decision · Program_Chairs · 2024-01-16

Reject